# Community Risk Factors in the COVID-19 Incidence and Mortality in Catalonia (Spain). A Population-Based Study

**DOI:** 10.3390/ijerph18073768

**Published:** 2021-04-04

**Authors:** Quim Zaldo-Aubanell, Ferran Campillo i López, Albert Bach, Isabel Serra, Joan Olivet-Vila, Marc Saez, David Pino, Roser Maneja

**Affiliations:** 1Environment and Human Health Laboratory (EH2 Lab), Forest Science and Technology Center of Catalonia, Ctra. de St. Llorenç de Morunys, km 2, 25280 Solsona, Spain; FCAMPILLO@hospiolot.cat (F.C.iL.); albert.bach@uab.cat (A.B.); roser.maneja@uab.cat (R.M.); 2Institute of Environmental Science and Technology (ICTA), Autonomous University of Barcelona (UAB), Z Building, ICTA-ICP, Carrer de les Columnes, UAB Campus, 08193 Bellaterra, Spain; 3Pediatric Environmental Health Specialty Unit, Pediatric Team of Garrotxa and Ripollès Regions, Olot and Garrotxa Region Hospital Foundation, 17800 Olot, Spain; 4Centre de Recerca Matemàtica, Edifici C, 08193 Bellaterra, Spain; iserra@crm.cat; 5Barcelona Supercomputing Center, 08034 Barcelona, Spain; 6Health Promotion Service in Girona, Agency of Public Health of Catalonia, Generalitat of Catalonia, 17003 Girona, Spain; jolivetv@gencat.cat; 7Research Group on Statistics, Econometrics and Health (GRECS), University of Girona, 17003 Girona, Spain; marc.saez@udg.edu; 8CIBER of Epidemiology and Public Health (CIBERESP), 28029 Madrid, Spain; 9Departament of Physics, Universitat Politècnica de Catalunya·BarcelonaTech, Esteve Terrades 5, 08034 Castelldefels, Spain; david.pino@upc.edu; 10Institut d’Estudis Espacials de Catalunya (IEEC-UPC), Gran Capità 2-4, 08034 Barcelona, Spain; 11Forest Science and Technology Center of Catalonia, Ctra. de St. Llorenç de Morunys, km 2, 25280 Solsona, Spain; 12Geography Department, Autonomous University of Barcelona (UAB), B Building, UAB Campus, 08193 Bellaterra, Spain

**Keywords:** COVID-19, air pollutants, cardiovascular diseases, psychological disorders, cancer, agri-food industry, land use and land cover data

## Abstract

The heterogenous distribution of both COVID-19 incidence and mortality in Catalonia (Spain) during the firsts moths of the pandemic suggests that differences in baseline risk factors across regions might play a relevant role in modulating the outcome of the pandemic. This paper investigates the associations between both COVID-19 incidence and mortality and air pollutant concentration levels, and screens the potential effect of the type of agri-food industry and the overall land use and cover (LULC) at area level. We used a main model with demographic, socioeconomic and comorbidity covariates highlighted in previous research as important predictors. This allowed us to take a glimpse of the independent effect of the explanatory variables when controlled for the main model covariates. Our findings are aligned with previous research showing that the baseline features of the regions in terms of general health status, pollutant concentration levels (here NO_2_ and PM_10_), type of agri-food industry, and type of land use and land cover have modulated the impact of COVID-19 at a regional scale. This study is among the first to explore the associations between COVID-19 and the type of agri-food industry and LULC data using a population-based approach. The results of this paper might serve as the basis to develop new research hypotheses using a more comprehensive approach, highlighting the inequalities of regions in terms of risk factors and their response to COVID-19, as well as fostering public policies towards more resilient and safer environments.

## 1. Introduction

The COVID-19 pandemic, caused by the Severe Acute Respiratory Syndrome Coronavirus 2 (SARS-CoV-2), has become a leading health concern worldwide. As of 31 May 2020, there were 5,939,234 confirmed cases and 367,255 deaths globally [1]. The severity and mortality have been related to aging and pre-existent health conditions, including respiratory and cardiovascular diseases, as well as psychological disorders and cancer [2,3]. Nevertheless, the geographic COVID-19 distribution within countries or regions has been uneven [4]. Socioeconomic status has also been pointed out as a community determining factor, but inconsistently for both richer and poorer populations [5,6]. In the same direction, inconclusive results have been found regarding population density [6,7,8]. Previous studies have reported the association between population physical distancing and COVID-19 spreading dynamics [9,10,11], as well as other weather conditions such as humidity and temperature [12]. These links might lie behind the local outbreaks of the pandemic in certain agri-food sectors such as meat and leather and fur industries [13,14]. However, other studies have recently pointed out that COVID-19 incidence correlates to ultraviolet radiation, rather than temperature-humidity [15,16].

Air pollution remains one of the main threats for human health worldwide and can also play a relevant role in the COVID-19 crisis mainly in two ways: increasing the severity of the virus’ clinical effects in chronically exposed populations and, probably to a lesser extent, promoting the virus’ airborne dispersion [17,18,19]. On one hand, according to the World Health Organization (WHO), there are 4.2 million deaths every year mostly due to cardiorespiratory diseases as a result of exposure to outdoor air pollution [20]. Recent studies have shown that ambient air pollution may be linked to the lethality of COVID-19 in Asia, Europe and America [21,22,23,24,25,26]. Thus, regions chronically exposed to nitrogen dioxide (NO_2_) and particulate matter (PM_2.5_ and PM_10_) seem to be more susceptible to the virus. Still, many of those studies do not include well identified health covariates [27,28,29] and are focused only on mortality. On the other hand, some authors have studied the role of particulate matter in the spreading of SARS-CoV-2 [12,30,31,32], principally in industrialised areas [33].

Air pollution and aerosol formation and distribution have been widely linked to Land Use and Land Cover (LULC) [34,35,36], with an especial concern regarding particulate matter (PM_2.5_ and PM_10_) [37,38,39]. In this sense, urbanised and industrial areas are associated with worse air quality than other LULC categories such as agricultural or forested areas [39]. LULC information is useful open source data which is associated with other factors like population density, biodiversity and economic activities [40], and has been identified as a suitable describer of the environment in studies relating the environment to human health [41]. For the aforementioned, research encompassing the associations between COVID-19 and LULC data appears to be relevant, since this spatial data (LULC) leverage socioeconomic and biophysical information of the environment.

In Catalonia (Spain), there was a heterogenous distribution of both COVID-19 incidence and mortality in the early stages of the pandemic. This suggests that differences in baseline risk factors across regions might have modulated the outcome of the pandemic. The purposes of this study are to:Analyse the associations between both COVID-19 incidence and mortality and long-term exposure to pollutant concentration (NO_2_ and PM_10_), while adjusting for demographic information, socioeconomic status and general health status (cardiovascular diseases, psychological disorders and all-cause cancer);Explore the potential links between agri-food industry and COVID-19 incidence and mortality as observed from the outbreaks in these particular industries;Screen, for the very first time, the potential use of the overall Land Use and Cover data on describing the geographical COVID-19 incidence and mortality.

## 2. Materials and Methods

### 2.1. COVID-19 Cases and Deaths

The number of patients infected with SARS-CoV-2 (cases) and deaths attributed to COVID-19 in Catalonia were gathered until 18 May 2020, after the first peak decreased and the incidence of new cases started to stabilise.

The number of cases was obtained from the RSAcovid19 records from the Catalan Health Department. We collected both the positive cases (patients positively diagnosed by a PCR—Polymerase Chain Reaction—or rapid diagnostic test) and suspicions cases (patients who presented symptoms compatible with COVID-19 and were classified as a possible case, even though they were diagnosed neither by a PCR nor by a rapid diagnostic test). All of them were active cases under the control of Epidemiologic Surveillance Service in Catalonia and were attributed to their residential Basic Health Area (BHA), the fundamental territorial unit through which Catalan Healthcare System is articulated and the unit of analysis of this paper. In total, 372 BHA compose the Catalan territory.

The number of registered deaths due to COVID-19 was obtained from the Catalan Agency for Health Quality and Evaluation (AQuAS) and the Central Register of Insured Persons of the Catalan Health Department. These data included not only people who were positively diagnosed by a laboratory test but also people who presented symptoms compatible with the illness. These open data are updated several times per day, so analyses and figures might change depending on the date. Furthermore, death observations might be modified by the Mortality Register of Catalonia once all death certificates have been collected [42].

Both data sets were provided already segmented by sex (male and female). Incidence and mortality rates were calculated using the number of cases and the number of deaths divided by the total amount of population within each BHA. Appendix A show the COVID-19 incidence and mortality rates, respectively (see Appendix A).

### 2.2. Comorbidities

During the first wave of the pandemic in Catalonia, COVID-19 tests were not conducted on every person showing symptoms. Rather, people with more severe symptoms or having pre-existent health conditions were more likely to be tested and thus, finally diagnosed. To control for the general health status of each BHA, we created three groups of principal health conditions explored by previous literature: cardiovascular diseases; psychological disorders and all-cause cancer. Pre-existent respiratory conditions could not be considered as the health dataset was incomplete.

The percentages of people presenting cardiovascular diseases (congestive heart failure, hypertension, ischemic cardiomyopathy and patients who suffered cerebrovascular accident), psychological disorders (depression, schizophrenia, intellectual disability, conduct disorder, attention deficit disorder and psychosis), and all-cause cancer were obtained from historical observational data from 2014 provided by the Catalan Health Department and the Catalan Agency for Health Quality and Evaluation (AQuAS). We lacked more recent data to control for the general health status of BHAs. However, the health outcomes assessed were prevalent illnesses with generally slight changes from one year to another. The data was aggregated by BHA and sex (male and female).

### 2.3. Demographic and Socioeconomic Data

Some authors have highlighted the prominent impacts of COVID-19 on elderly people, especially in nursing homes [43]. Others have also focused their studies in the importance of sex [44]. We controlled for sex and elderly people by calculating the percentage of people over the age of sixty-five in each BHA and distinguishing the COVID-19 cases and deaths between males and females. In addition, socioeconomic data were extracted from the Catalan Health Observatory. We used the Composed Socioeconomic Index (CSI) [45], that is calculated for each BHA. This index is used in the assessment of resources for Primary Health, which includes a set of socioeconomic variables: economic income, education, professional occupation, life expectancy, premature death rate and preventable hospitalizations rate. This is a continuous variable measured from 0 to 100 (0 being the poorest and 100 the richest). Previous works have suggested dividing such data into septiles [3]. However, after testing the model, we opted for using quintiles from a very low (E; CSI ≥ 0 and <20) to a very high (A; CSI ≥ 80) socioeconomic status (SES).

### 2.4. Air Pollution

Long-term exposure to air pollutants was assessed using the modelling of the NO_2_ and PM_10_ annual average (µg/m^3^) in Catalonia, corresponding to the 2016 assessment from the General Direction of Environmental Quality and Climate Change of the Catalan Government.

We calculated the annual weighted average for each BHA through GRASS GIS (GRASS Development Team, 2017. Geographic Resources Analysis Support System (GRASS) Software, Version 7.2. Open Source Geospatial Foundation. Electronic document: http://grass.osgeo.org (accessed on 23 May 2020)) (see Appendix A showing the annual weighed average of NO_2_ and PM_10_ (µg/m^3^) for each BHA (2016)).

Besides air pollution data from 2016, we created a dataset for the period 2018–2019 (the most up-to-date period with data available). We combined three data sources (pollution data from the Catalan Government; Smart Citizen, a citizen science project from the European Community’s H2020; and pollution data from the European Environment Agency). Then, we calculated the annual average for each pollutant in each BHA containing sensors, which yielded 63 BHAs with values for NO_2_ and 91 with values for PM_10_. After controlling for possible differences between both periods (2016 and 2018/2019) and finding no significant differences, we chose the modelling of the NO_2_ and PM_10_ annual average for 2016 because it provided information for all Catalonia. Results of the two independent t-tests assessing significant differences between pollutant concentration levels (NO_2_ and PM_10_) in 2016 and in 2018/2019 are provided in the Results section.

Other air pollutants have been widely used to assess pollution levels. Previous research hypothesised that long-term exposure to O_3_ and PM_2.5_ adversely affects the respiratory and cardiovascular systems, increasing mortality risk and also exacerbating the severity of COVID-19, worsening the prognosis of the disease [46,47]. In this sense, O_3_ levels has been found to be associated with COVID-19 confirmed cases [48] and PM_2.5_ to be a highly significant predictor of the number of confirmed COVID-19 cases, deaths and hospital admissions [48,49]. Although assessment of the independent effect of the abovementioned pollutants would have been of interest, we only used NO_2_ and PM_10_ data, as they were provided for all Catalan territory.

### 2.5. Agri-Food Industry

Agri-food industry geographic information was extracted from Catalan Agri-food industry Records (http://agricultura.gencat.cat/ca/serveis/registres-oficials/agroalimentacio/registre-industries-agraries-alimentaries-catalunya/ (accessed on 1 June 2020)). The industries are classified depending on their industrial sector: slaughter of livestock, conservation and elaboration of meat products; preparation and conservation of fish, crustaceans and molluscs; preparation and preservation of fruits and vegetables; manufacturing of vegetables and animal oils and fats; manufacturing of milk products; manufacturing of grain mill products, starches and starch products; manufacturing of bakery and pasta products; manufacturing of other food products; manufacturing of products for animal feeding; manufacturing of beverages; forest industries; and other agricultural industries.

We split the category “other agricultural industries” into two main subtypes: “Leather and fur industry” (industries based on preparation, tanning and dyeing animal skins) and “Garden industry” (industries based on seed conditioning and handling, substrate production and ornamental plant conservation), as we considered that these two sectors were poorly represented in the above classification. The total number of industries of each type was collected within each BHA.

### 2.6. Land Use and Land Cover Data

To describe the environment of each BHA we used the most updated and detailed Land Use and Land Cover data of Catalonia, the Land Use and Cover map for 2017. This is a tool generated with automated image classification of a 30-m resolution. The images are obtained thought Landsat satellite (Landsat-5, Landsat-7, Landsat-8 and Sentinel-2) using both their sensors Thematic Mapper (TM), Enhanced Thematic Mapper Plus (ETM+), Operational Land Imager (OLI) and Multispectral Imager (MSI), and complementary information such as the Urban Map of Catalonia and the graph of the Catalonia infrastructures network. It also incorporates the cartographic database of forest fires from the Ministry of Agriculture, Livestock, Fisheries and Food of Catalonia, and the LIDAR database from the Institut Cartogràfic i Geològic de Catalunya (ICGC) (http://territori.gencat.cat/ca/01_departament/12_cartografia_i_toponimia/bases_cartografiques/medi_ambient_i_sostenibilitat/bases_miramon/territori/mapa-dusos-i-cobertes-del-sol/index.html (accessed on 30 May 2020))

As Table 1 shows, we reclassified the 25 Land Use and Land Cover (LULC) categories into four broader categories: urban areas; industrial, commercial and transport units; agricultural areas; and forest and semi-natural areas. In this classification, categories referring to water bodies (inland and marine waters) and bare land were not considered due to their low significance.

When proportions of land use and cover composing geographical regions are analysed, each observation is a vector of proportions of specific LULC categories [50]. This characteristic raises the problem of singularity (a constant sum constraint) as the vectors (also called compositions) describe the relative contribution of each part (the components) on the whole. So the information is present in the ratios of the components rather than in each component [51,52,53]. Following Müller et al. (2018) [54], we avoided the singularity constraint by applying an isometric logratio (ilr) transformation to the four LULC variables. This transformation moves the compositions isometrically from the simplex with the Aitchison geometry to the standard real space with the Euclidean one [53]. As recommended [54], we used a Log2 transformation, as it facilitated the understanding of the estimates. With this transformation, a unit additive increment in the ilr-transformed variable is equal to a two-fold multiplicative increase in the relative dominance of the original composition variable x, as a base-2 logarithm is used. In other words, this means that the relative dominance of a specific LULC category is doubled in comparison to the geometric mean of all the rest LULC variables [54].

### 2.7. Statistical Analysis

To assess the associations between COVID-19 incidence and mortality and the explanatory variables, we fitted a generalised linear model, in the binomial family, with a logit link. This model fit was selected as the dependent variable followed a Binomial distribution.
$$Y_{i}\sim Bernoulli\left(p_{i}\right)\;\mathrm{for}\;i=1,\;\ldots ,\;n.$$
$$Logit\left({µ}_{i}\right)=\log\left(\frac{p_{i}}{1-p_{i}}\right)=\beta_{0}+\overset{n}{\underset{i=1}{\sum }}\beta_{i}\times X_{i},$$
where $Y_{i}$ was the binary (Bernoulli) response variable; $p_{i}$ was the probability of successes $P\left(Y_{i}=1\right)$, in this case, 1 stands for a confirmed COVID-19 case or death; *µ_i_* is the expected value of each $Y_{i}$ which is equal to the probability of successes $p_{i}$; $\beta_{0}$ is the intercept, and $\beta_{i}$ denotes the logistic regression coefficients for the design matrix X of covariables i. 

Logistic regression analyses with 95% Wald confidence intervals (95% CI) were performed to assess the association between both incidence and mortality rate of COVID-19 (number of confirmed COVID-19 cases or deaths within a given BHA/total number of people living within such BHA) and the rest of covariates, while adjusting for demographics, socioeconomic and comorbidity covariates. The model was fitted using population size of each BHA as weights. We built a main model using the demographics, socioeconomic and comorbidity covariates and then, human activity covariates, as well as land use and cover covariates, were included in the model separately (see Table 2).

Statistical analysis were conducted using the R language environment for statistical computing, R version 3.6.2 (12 December 2019) [55].

## 3. Results

Homogeneity of groups in terms of pollutant concentration levels was assessed using two independent t-tests (Table 3) for the specific BHA which we had available information (63 BHA, for NO_2_; and 91 BHA, for PM_10_). Based on the t-tests outcomes, no significant differences were noted between the annual average of pollutants in 2016 and in 2018/2019 for neither pollutant (NO_2_; *t* = 0.792, *p* = 0.428, and PM_10_; *t* = −1.559, *p* = 0.119).

The adjusted odds ratio (OR) with 95% confidence intervals for the association between COVID-19 incidence and mortality and the explored covariates are shown in Table 4 and also represented in Appendix A.

In the main model using demographic, socioeconomic and comorbidity covariables, BHAs with more percentage of people aged above 65 years, of A (very high) and B (high) socioeconomic status (SES) showed a positive association with both COVID-19 incidence and mortality. In these cases, estimates for mortality were greater than for incidence. Contrarily, BHAs of D (low) and E (very low) SES were associated with decreased levels of COVID-19 incidence and mortality. However, when tested alone (without adjusting for the rest of covariates), they showed a non-significant effect.

All three comorbidity variables were positively associated with both COVID-19 incidence (OR 1.003 95% 1.0020–1.0049 for cardiovascular diseases; OR 1.148 95% 1.1418–1.1545 for psychological disorders; and OR 1.021 95% 1.0153–1.0258 for all-cause cancer) and mortality (OR 1.007 95% 1.0006–1.0136 for cardiovascular diseases; OR 1.312 95% 1.2809–1.3435 for psychological disorders; and OR 1.102 95% 1.0774–1.1272 for all-cause cancer). Again, the estimates for mortality were found higher than for incidence in all three comorbidity variables.

Finally, sex (comparing females to males) showed a positive significant effect on the incidence of COVID-19 (OR 1.772 95% 1.7577–1.7870) and a non-significant effect on the mortality (OR 1.034 95% 0.9974–1.0724). It also showed a non-significant effect on COVID-19 mortality when tested unadjusted.

We found a positive association between COVID-19 mortality and the annual average of both pollutants (NO_2_ and PM_10_). Our model showed that, when the rest of covariates held constant, an increase of 10 µg/m^3^ in NO_2_ and PM_10_ annual average multiplied the odds of COVID-19 mortality by 1.138 (95% 1.1245–1.162) and by 1.598 (95% 1.5104–1.6936), respectively. Regarding COVID-19 incidence, PM_10_ also showed a positive association with COVID-19 incidence (OR 1.003 95% 1.0015–1.0038), while NO_2_ showed a negative association when tested adjusted for the rest of covariates (OR 0.999 95% 0.9989–0.9996).

As to the type of agri-food industries, we found several types that showed a reduced risk of both COVID-19 incidence and mortality (fish industry, vegetable, animal oils and fats, grain mill, bakery, other food products, animal feeding, beverage industry and garden industry). Milk products showed a non-significant effect on COVID-19 incidence and a negative effect on COVID-19 mortality. In addition, meat and forest industry showed a positive effect on the incidence of COVID-19 (OR 1.002 95% 1.0012–1.0019 for meat industry and OR 1.004 95% 1.0011–1.0077 for forest industry) but a negative effect on the mortality (OR 0.995 95% 0.9926–0.9965 for meat industry and OR 0.945 95% 0.9278–0.9632 for forest industry. However, unlike forest industry, meat industry showed a positive significant effect when tested unadjusted, as well. Finally, leather and fur industry were the only type of agri-food industry that were associated with increased levels of both COVID-19 incidence (OR 1.070 95% 1.0624–1.0779) and of COVID-19 mortality (OR 1.110 95% 1.0776–1.1441).

Regarding LULC data, we found a decreased risk of COVID-19 incidence for ilr-Industrial areas and ilr-Agricultural areas. In other words, when the relative dominance of industrial areas and agricultural areas were doubled in a given BHA with respect to the rest of LULC categories, the odds for COVID-19 incidence was expected to be reduced by a 0.010% (95% 0.0079–0.0116) and 0.018 % (95% 0.0165–0.0194), respectively. On the other hand, for ilr-Urban areas and ilr-Forested areas the odds for COVID-19 incidence was expected to be increased by 0.006% (95% 0.0048–0.0076) and 0.014% (95% 0.0131–0.0158), respectively. As for the COVID-19 mortality, ilr-Urban and ilr-Industrial areas showed positive significant effects (OR 1.050 95% 1.0440–1.0569, and OR 1.039 95% 1.0304–1.0477, respectively), while ilr-Agricultural and ilr-Forested areas showed negative significant effects (OR 0.936 95% 0.9303–0.9422 and OR 0.991 95% 0.9856–0.9971, respectively)

### Main Model Adjustment

For illustrative purposes, the main model adjustment is shown for COVID-19 cases and deaths instead of the incidence and mortality rate. We noted no important differences between the expected values for males and for females for the main model. Thus, we assessed the model with the total number of cases and deaths (females + males).

Figure 1 shows a scatter plot were the observed number of COVID-19 cases (on the left) and deaths (on the right) are plotted against the expected number of COVID-19 cases and deaths predicted by the model. Those BHA which fulfilled the criterion that the difference between the observed rate and the fitted rate was either >0.03 or <0.03 (for COVID-19 cases), and >0.004 or <0.004 (for COVID-19 deaths) were identified as outliers.

The outliers coincide with either northern BHAs with high amounts of forest and semi-natural areas, low population and high incidence and mortality cases, or with regions from Central Catalonia where incidence and mortality were also high (“Barcelona 05D”, “Girona-4”, “Alt Berguedà”, “la Pobla de Segur”, “Sant Quirze de Besora” and “Igualada-2” for COVID-19 cases, Figure 1 left; and “Cardona”, “Alt Berguedà”, “Capellades”, “Vilanova del Camí” and “Igualada-2” for COVID-19 deaths, Figure 1 right). As a matter of fact, two of the observed outliers (“Vilanova del Camí” and “Igualada-2”) were BHAs in which the early outbreaks of the pandemic occurred.

Additionally, Figure 2 and Figure 3 show the observed number of COVID-19 cases and deaths (on the left) and the expected number of cases and deaths (on the right) for each BHA predicted by the main model. In purple, there are represented those BHAs where the expected value was overestimated (difference between observed cases or deaths and expected cases or deaths < Q1) by the main model. On the other hand, in orange there are represented those BHAs where the expected value was underestimated (difference > Q3) by the main model. In green, those BHAs where the difference between the observed value and the expected fell within the Q1 and the Q3 are plotted.

## 4. Discussion

This cross-sectional study aimed to evaluate the associations between COVID-19 incidence and mortality and long-term exposition to air pollution (NO_2_ and PM_10_) while adjusting for demographic (sex, percentage of people aged above 65 years), socioeconomic (quintile division of the Composed Socioeconomic Index) and comorbidity data (percentage of people presenting cardiovascular disease, psychological disorders and all-cause cancer). Additionally, for the first time, the contribution of agri-food industry type and the overall Land Use and Land Cover data was also explored to explain the geographical distribution of COVID-19 incidence and mortality, leading to novel results. 

### 4.1. Demographics

Registered cases of COVID-19 in Catalonia have a clear female predominance (165,597 cases in females compared to 95,317 cases in males). Compared to other nations, the proportion of women in the incidence rate is only surpassed by Wales (63.46% vs. 64.18%), while being still slightly higher than the Netherlands (62.45%), Scotland (62.01%), Northern Ireland (61.94%), or Sweden (59.37%) [56]. Mortality was also higher among females (50.41%), but below what has occurred in Finland (52.00%) and the Republic of Ireland (50.50%) [56]. Catalonia has a positive small prevalence of female population (50.9%). In addition, this predominance positively increases for people older than 65 years (57.0%), while being reversed in 0–24-year-old children (around 48.6%) [57]. With older people being the most affected by COVID-19 and the younger the least (in the early stages of the pandemic), women might be expected to carry most of the burden. In addition, research has highlighted women as composing the majority of the healthcare workforce in the US, and also with roles requiring more close and prolonged contact with patients [58]. Furthermore, for employed women or single parents, gender disparities may even be accentuated, as women are disproportionally responsible for the bulk of domestic tasks, including not only childcare but also eldercare [59]. These factors might explain our results showing women having 77.2% more risk of COVID-19 infection than males.

However, other countries with comparable age-gender pyramids (younger male population and older female population), such as Italy or the United States [60], have not experienced this phenomenon, following the global trend of male predominance [61,62].

However, we did not find greater risk of COVID-19 mortality for females, as the number of deaths for females and males was not significantly different (6098 and 5998, respectively).

Recent studies have pointed out that older age is as a major individual risk factor for severity of the COVID-19 infection and mortality [58,63]. We detected this effect in the adjusted and non-adjusted models for both COVID-19 infection and mortality. Nevertheless, the effect of age was reduced when adjusted for the rest of covariates.

### 4.2. Socioeconomics

Previous studies have suggested that socioeconomically deprived groups were associated with a higher risk of confirmed COVID-19 infection [64]. At the beginning of the outbreak, some authors suggested that working class people might be more exposed to the virus, as they were associated with the use of public transport [65]. However, other reports encouraged its use as the incidence of COVID-19 attributed to public appeared to be very low [66], even though safety countermeasures should be taken into account [67]. Regarding deprived people, some authors suggest that this group might face several disadvantages which make physical distancing a difficult issue [68]. That is, besides showing greater mobility due to the impossibility of working from home, lower-income population might tend to visit denser places (grocery stores, religious establishments, etc.), and spend longer times than upper class populations [69]. In Catalonia, some studies observed higher incidence of COVID-19 in poorer areas of Barcelona city [70].

Despite all the research showing a greater impact of COVID-19 on lower SES classes, our results seem to point to the other way around. We found higher incidence and mortality ratios for higher SES BHAs compared to medium SES. This effect was significant before and after adjusting for the rest of the covariates. In addition, although a non-significant effect was found for low (D) and very low (E) SES BHAs when tested unadjusted, when adjusting them into the model, they showed a significant negative association with both COVID-19 incidence and mortality. It is possible that differences between SES classes in Catalonia were not as noticeable as they were in other regions (in the UK, for example [64]). However, it is also possible that the Composed Socioeconomic Index used to measure the SES at area level might be weak measurement to detect individual-based characteristics. Nevertheless, as shown elsewhere [69], using a more detailed unit of analysis (e.g., census area) or completing SES information with individual-based information [64] might result in better estimations as to the impact of SES on COVID-19 incidence and mortality.

### 4.3. Comorbidities

Chronic medical conditions have been linked to disproportionate morbidity due to SARS-CoV-2 virus [58]. Regarding previous literature on SARS-CoV, some authors have reported that cardiovascular comorbidities might be the most important components for predicting adverse outcome, increasing the risk of death by twice as much as other risk factors [71]. In a recent meta-analysis [72], the proportion of cardia-cerebrovascular disease in patients with COVID-19 was found to be 16.4%. A proportion much higher than what is found in the general population [72]. In this sense, many researchers acknowledge the consistent association between cardiovascular disease and SARS-CoV-2 [2,73,74,75].

In another sense, some researchers have reported that people diagnosed with psychological disorders had significantly higher odds of COVID-19 infection than people without a psychological disorder, with the strongest effect for depression and schizophrenia [76]. In the same way, these authors reported that the death rate for patients with both a recent diagnosis of psychological disorder and COVID-19 infection was higher than patients with COVID-19 infection but with no psychological disorder [76].

Other research also states the role of cancer in aggravating the prognostics of COVID-19 [73]. In this regard, people with ongoing cancer treatments have shown higher risk because their immune system is compromised [77]. 

Our results are aligned with previous literature showing increased risk for both COVID-19 infection and mortality for those areas with more percent of people suffering from cardiovascular disease, psychological disorders and all-cause cancer. Similar to previous literature, we used these variables to control for the general health status of the BHAs, building our main model. They all showed a positive significant association with both COVID-19 infection and mortality before and after adjustment. This research adds evidence that these comorbidity variables are significant predictors.

Additionally, other relevant comorbidities such as obesity [78] or respiratory illnesses (e.g., COPD [79] and asthma [2]) have also been found to be positively associated with both infection and mortality for COVID-19. Our study was not able to control for these variables as we lacked the information. However, future studies might also use respiratory illnesses to describe the general health statutes of the unit of analysis.

### 4.4. Air Pollution

The major route of transmission for COVID-19 is through small droplets and aerosols of different sizes exhaled by an infected person when breathing, talking, coughing or sneezing [29,80,81]. Additionally, some research suggests the rapid spread of the SARS-CoV-2 could be explained by air pollution-to-human transmission (e.g., airborne transmission) [17,18,19]. Considering that the data used in this paper was historical (2016), we could not assess the relationship between short-term exposition to high levels of air pollutants (e.g., PM_10_) and the COVID-19 incidence or mortality and hence, provide evidence neither supporting these hypotheses nor against them.

In our opinion, the principal pathway linking air pollution to increased levels of COVID-19 incidence and mortality is the worse health status of more exposed populations [29,82,83].

Long-term exposure to air pollution has been widely linked to cardiovascular diseases, respiratory illnesses, psychological disorders and cancer [84,85,86]. We believe that this might explain the association between more polluted areas and more severe and lethal forms of COVID-19 [26,80]. In this sense, areas more chronically exposed to higher air pollution levels would presumably be in worse health status and thus, showing increasing levels of COVID-19 mortality. Regarding the incidence of the virus, the positive association between increased pollutant levels and increased incidence levels of COVID-19 (at least for PM_10_) would be explained as during the early stages of the pandemic, people with pre-existent health conditions, or with more severe symptoms, were more likely to be tested, and thus, to finally be diagnosed as a new case.

As shown elsewhere [87], NO_2_ and PM_10_ effects on COVID-19 mortality remained significant after adjusting for socioeconomic, demographic and health-related variables. When adjusted in the model, NO_2_ showed a negative association with COVID-19 infection levels. In this sense, other relevant research conducted in Catalonia [88] highlights an association between NO_2_ and COVID-19 incidence, but the association was only found in more polluted BHAs. Our approach of using this data for all Catalonia without stratifying for more polluted areas might prevent us from detecting the aforementioned effect.

### 4.5. Forest, Meat, and Leather and Fur Industry

Our results show a significant positive effect of forested areas on COVID-19 incidence. Although forest industries might apparently be more abundant in BHAs with more forested areas, its positive effect was only found when it was adjusted, showing a significant negative effect when tested alone. We hypothesise that, rather than a positive independent effect for forest industry on COVID-19 incidence, possible associations with some main model covariates might have contributed to changing the direction of the effect.

In Catalonia, there is a huge production of pork meat, with a degree of self-sufficiency of 228.73%, that has been constantly growing in recent decades [89]. While swine breeding is concentrated in Lleida region and Central Catalonia, most slaughterhouses and pork meat industries are located in Central Catalonia and Girona region [89]. Working conditions in slaughterhouses and meat industries such as low temperatures, high humidity, overcrowding, physical effort and other things may contribute to amplifying virus viability and transmission [90]. These conditions might also be found in other types of industries with high working density, making them prone-to-infection industries. However, apart from forest industries, we only found animal-related industries, namely the meat industry and the leather and fur industry, to be related with COVID-19 incidence and mortality (only the leather and fur industry).

In other coronavirus infections such as MERS, there was a high prevalence of infection in slaughterhouse workers compared to the general population [91]. COVID-19 transmission has been reported in the meat and poultry industry [13] and slaughterhouses are now considered a new front line in the COVID-19 pandemic [92]. In the same direction, local outbreaks in the fur industry have also been reported, particularly in the mink furriery [93,94]. The fact that this particular economic activity is significantly increasing both the incidence and the mortality rate in our model makes it plausible that this kind of industry poses a unique and independent risk for COVID-19 transmission.

In a recent study from the Netherlands [94], the authors reported that minks are susceptible for SARS-CoV-2. In addition, that infected animals are able to transmit the virus among each other. The authors also claim that although mink farms are present in other countries in Europe, China and the US, only the Netherlands has reported SARS-CoV-2 infections in these animals. In our study, we did not identify the animal species of the leather and fur industries we assessed. However, and given the results shown, more attention and research should be placed upon this specific industry.

In this sense, it is advised that COVID-19 pandemic should trigger a profound transformation of industrial animal agriculture by improving living conditions and increasing their space through extensive farming, diversify the protein source industry to increase far more sustainable plant-based market shares, and empowering the ecological transition of animal farmers [95].

### 4.6. Land Use and Cover

LULC data has been shown to be a suitable describer for the environment surrounding individuals in studies linking the environment to human health [41]. Unlike other environmental data sets, they combine both the biophysical (e.g., temperature, humidity, soil features) and socioeconomic (e.g., political, economic, cultural) drivers of a territory [40,96]. Given the uneven geographical distribution of the virus in Catalonia, we wanted to screen whether environmental composition of the BHAs (seen as urban, industrial, agricultural and forested areas) might be related with the impact of COVID-19.

Urban areas and industrial, commercial and transport units are known to be more associated with air pollution, aerosol emission, human mobility and higher population density [97]. These factors might be the reasons behind the increased risk of COVID-19 mortality shown by the two ilr-transformed LUC categories, and an increased risk of COVID-19 infection for urban areas. Industrial areas showed a negative association with the incidence of COVID-19. This suggests that rather than the extension of the LULC category, the type of industry might be more relevant (as appreciated for agri-food industries).

On the other hand, agricultural areas and forested areas are more related to better air quality [34,38,98,99], which might lead to higher general health status. That, in turn, might explain the negative association for both categories with COVID-19 mortality. However, despite people remaining under lockdown during most of the period analysed in this paper, agricultural tasks were considered essential services. These tasks mainly include individual work and are frequently done outdoors. Additionally, agricultural areas tend to be less populated which increases social distancing. We hypothesise that these aspects might have prevented regions with higher agricultural areas to easily register COVID-19 cases.

Although forest and semi-natural areas showed a decreased risk of COVID-19 mortality, the increased risk for COVID-19 incidence was somewhat a surprising result. Forested areas are widely known for their air purification role [38]. Furthermore, vegetation can also lessen other determinant variables for aerosol dispersion such as wind speed. In the same direction, areas with an increased amount of forest are associated with less population density and hence, more physical distancing. In Catalonia, the BAHs with the highest amount of forest and semi-natural areas tend to be sparsely populated. Moreover, many of these BAHs held second residences, mainly belonging to people living in the Metropolitan Area of Barcelona, who may have commuted to the countryside as soon as the emergency state was declared [100]. In these regions, few cases can be translated into high incidence rates, which might explain the increased risk of COVID-19 infection for higher levels of forested areas.

During the first wave of the pandemic, people remained at home, decreasing human interactions. In future studies, LULC data might be leveraged encompassing variables such as population density, air quality, biodiversity and economic activities to further validate LULC data in scenarios with mobile people.

### 4.7. Limitations

We implemented a cross-sectional design, so we could not escape from many of the limitations of ecological regression analysis highlighted elsewhere [46]. One of the major constraints is that, when using these designs, causal inference cannot be spotted. Nevertheless, these studies do leverage data for an entire population (Catalonia in our case) and are able to make conclusions at the area level (e.g., BHAs), which might be useful for policy-making [46]. Furthermore, the associations detected in this paper can provide justification for ongoing or future research.

We did not study the evolution of the epidemic taking place later than the 18 May 2020. As for age groups, we only controlled for the percentage of elder people (>65 years). However, the advance of the epidemic has shown that many other age groups are vulnerable and should be considered in further analyses.

Controlling for other pre-existent health conditions such as obesity or respiratory illnesses and incorporating a greater variety of human mobility data (in scenarios with mobile people) such as the public transport network may enhance future research.

Although we controlled for significant differences for the pollutant concentration levels between 2016 and 2018/2019, accounting for the most recent modelling of the NO_2_ and PM_10_ annual average (µg/m^3^) in Catalonia may have improved the analysis as well. Furthermore, controlling for other air pollutants such as O_3_ or PM_2.5_, which have been described as relevant in previous research, might enhance future research.

In the same direction, the incapability for acquiring more updated data led us to use different datasets from different years for all the assessed covariates. However, the estimations found are consistent with previous research, which adds evidence as to the independent effect of the covariates assessed.

## 5. Conclusions

Recent literature has highlighted the importance of controlling for covariates in studies linking air pollution to COVID-19. We used a main model with demographic, socioeconomic and comorbidity covariates highlighted from previous research as important predictors. This allowed us to take a glimpse of the independent effect of each explanatory variable when controlled for the main model covariates. Our findings are aligned with previous research showing that the baseline features of the regions in terms of health status, pollutant concentration levels (NO_2_ and PM_10_), type of agri-food industry and type of land use and land cover have modulated the impact of the COVID-19 at a regional scale. A warning is made regarding future pandemics caused by respiratory infectious diseases. Thus, actions that improve air quality, diversify economic activities and enhance overall public health should be considered, not only to weaken the intensity of the current coronavirus, but for other virus-related problems expected to come.

## Figures and Tables

**Figure 1 ijerph-18-03768-f001:**
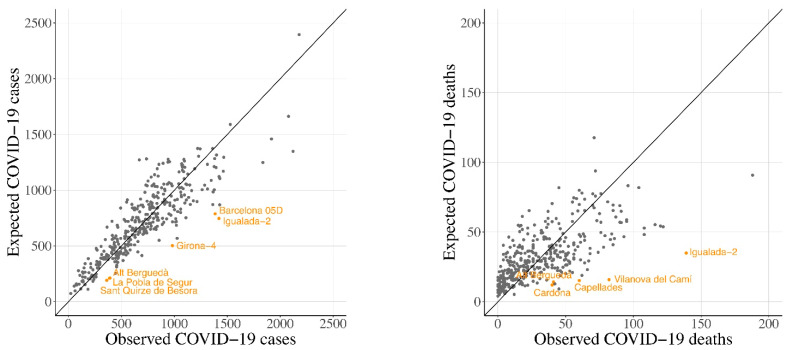
Scatter plot of the observed number of COVID-19 cases (**left**) and deaths (**right**), and the expected value predicted by the main model, logarithmic transformation has been performed.

**Figure 2 ijerph-18-03768-f002:**
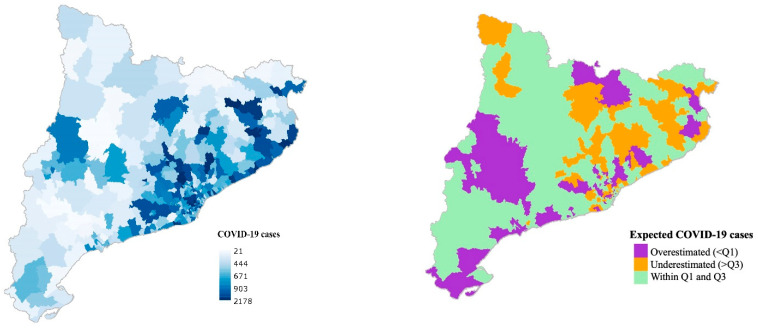
Number of observed COVID-19 cases (**left**) and the quartile distribution of the number of expected COVID-19 cases predicted by the main model (**right**).

**Figure 3 ijerph-18-03768-f003:**
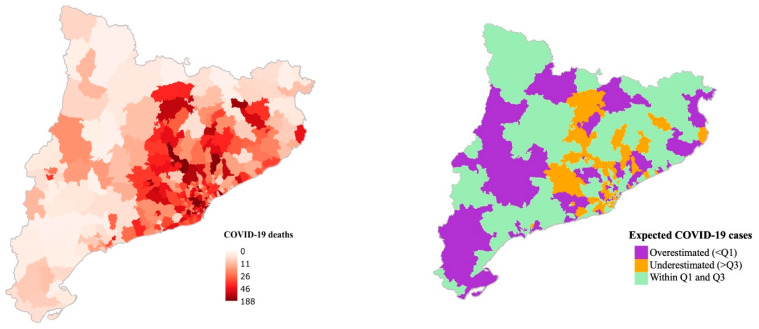
Number of observed COVID-19 deaths (**left**) and the quartile distribution of the number of expected COVID-19 deaths predicted by the main model (**right**).

**Table 1 ijerph-18-03768-t001:** Reclassification of the 25 LULC categories of the Land Use and Cover Map of Catalonia (2017) into four broader categories.

Urban Areas	Industrial, Commercial and Transport Units	Agricultural Areas	Forest and Semi-NATURAL Areas
Discontinuous urban fabric	Industrial or commercial units	Permanently irrigated land	Lowland natural grasslands
Continuous urban fabric	Road and rail networks and associated land	Non-irrigated arable land	Montane natural grasslands
		Unirrigated Fruit tress	Alpine natural grasslands
		Irrigated Fruit trees	Transitional woodland/shrub
		Vineyards	Wetland vegetation
		Rice fields	Coniferous forest
		Citrus trees	Broad-leaved forest
			Sclerophyll forest

**Table 2 ijerph-18-03768-t002:** Covariates tested in the model. All the variables were calculated within each BHA, the unit of analysis.

Covariate (Units)	Description
Demographics, socioeconomic status, and comorbidity (Main model)	
Sex: Females	Categorical variable comparing females to males, used as a reference level.
Percent > 65 (%)	Percentage of people aged above 65 years.
SES A	Socioeconomic status categorised with 5 levels, comparing very high, high, low and very low (A, B, D, E) socioeconomic status to normal (C), used as the reference level. Data from 2014.
SES B	
SES D	
SES E	
Cardiovascular diseases (%)	Group variable. Percentage of people with congestive heart failure, hypertension, ischemic cardiomyopathy or who suffered cerebrovascular accident in 2014.
Psychological disorders (%)	Group variable. Percentage of people with depression, schizophrenia, intellectual disability, conduct disorder, attention deficit disorder or psychosis in 2014.
All-cause cancer (%)	Group variable. Percentage of people with any type of cancer in 2014.
Human activity	
NO_2_ (µg/m^3^) *	Nitrogen dioxide annual weighed average in 2016.
PM_10_ (µg/m^3^) *	Particulate matter with diameter of 10 µm annual weighed average in 2016.
Meat industry *	Number of industries based on slaughtering of livestock, conservation and elaboration of meat products in 2020.
Fish industry *	Number of industries based on preparation and conservation of fish, crustaceans and molluscs in 2020.
Vegetable industry *	Number of industries based on preparation and preservation of fruits and vegetables in 2020.
Animal oils and fats *	Number of industries based on manufacturing of vegetable and animal oils and fats in 2020.
Milk products *	Number of industries based on manufacturing of milk products in 2020.
Grain mill industry *	Number of industries based on manufacturing of grain mill products, starches and starch products in 2020.
Bakery industry *	Number of industries based on manufacturing of bakery and pasta products in 2020.
Other food products *	Number of industries based on manufacturing of other food products in 2020.
Animal feeding *	Number of industries based on manufacturing of products for animal feeding in 2020.
Beverage industry *	Number of industries based on manufacturing of beverages in 2020.
Forest industry *	Number of forest industries in 2020.
Leather and fur industry *	Number of industries based on preparation, tanning and dyeing animal skins in 2020.
Garden industry *	Number of industries based on seed conditioning and handling, substrate production and ornamental plant conservation in 2020.
Land use and Land cover	
ilr-Urban areas *	Isometric logratio (ilr) transformation of the percentage of urban areas in a given BHA. Numerical variable.
ilr-Industrial areas *	Isometric logratio (ilr) transformation of the percentage of industrial, commercial and transport unit areas in a given BHA. Numerical variable.
ilr-Agricultural areas *	Isometric logratio (ilr) transformation of the percentage of agricultural areas in a given BHA. Numerical variable.
ilr-Forested areas *	Isometric logratio (ilr) transformation of the percentage of forested and semi-natural areas in a given BHA. Numerical variable.

* Variables were included in the model separately.

**Table 3 ijerph-18-03768-t003:** Independent t-tests between mean pollutant concentration levels in 2016 and in 2018/2019.

	Mean ± SD		Statistical Results		
Variables	2016 Concentration Levels	2018/2019 Concentration Levels	df	*t*	*p*-Value
NO_2_	20.23 ± 12.163	21.37 ± 10.700	246	0.792	0.428
PM_10_	21.52 ± 4.397	20.72 ± 5.241	351.37	−1.559	0.119

**Table 4 ijerph-18-03768-t004:** Associations between COVID-19 incidence and mortality and the rest of covariates. The main model controlled for demographics, socioeconomics and comorbidity covariables. Human activity covariates as well as land use and cover covariates were included in the model separately.

	Incidence of COVID-19				Mortality of COVID-19			
	Adjusted Main Model		Unadjusted		Adjusted Main Model		Unadjusted	
Covariates	Odds Ratio (95% CI)	*p*-Value	Odds Ratio (95% CI)	*p*-Value	Odds Ratio (95% CI)	*p*-Value	Odds Ratio (95% CI)	*p*-Value
Main Model								
Sex: Female	1.772 (1.7577–1.7870)	***	1.723 (1.7087–1.7366)	***	1.034 (0.9974–1.0724)	-	0.990 (0.9551–1.0257)	-
Percent > 65	1.006 (1.0047–1.0072)	***	1.018 (1.0171–1.0189)	***	1.023 (1.0171–1.0281)	***	1.052 (1.0481–1.0562)	***
SES A (very high)	1.199 (1.1832–1.2150)	***	1.171 (1.1568–1.1848)	***	1.547 (1.4556–1.6434)	***	1.523 (1.4414–1.6093)	***
SES B (high)	1.126 (1.1116–1.1402)	***	1.153 (1.1387–1.1674)	***	1.241 (1.1696–1.3166)	***	1.346 (1.2702–1.4271)	***
SES D (low)	0.967 (0.9542–0.9800)	***	0.998 (0.9849–1.0114)	-	0.914 (0.8573–0.9754)	*	1.015 (0.9517–1.0815)	-
SES E (very low)	0.956 (0.9432–0.9688)	***	0.994 (0.9806–1.0067)	-	0.908 (0.8511–0.9677)	**	1.011 (0.9493–1.0778)	-
Cardiovascular diseases	1.003 (1.0020–1.0049)	***	1.016 (1.0153–1.0173)	***	1.007 (1.0006–1.0136)	*	1.038 (1.0336–1.0423)	***
Psychological disorders	1.148 (1.1418–1.1545)	***	1.057 (1.0517–1.0627)	***	1.312 (1.2809–1.3435)	***	1.255 (1.2282–1.2827)	***
All-cause cancer	1.021 (1.0153–1.0258)	***	1.084 (1.0805–1.0883)	***	1.102 (1.0774–1.1272)	***	1.239 (1.2205–1.2584)	***
Human activity								
NO_2_	0.999 (0.9989–0.9996)	***	1.002 (1.0014–1.0020)	***	1.013 (1.0118–1.0151)	***	1.017 (1.0154–1.0182)	***
PM_10_	1.003 (1.0015–1.0038)	***	1.009 (1.0077–1.0098)	***	1.048 (1.0421–1.0541)	***	1.050 (1.0451–1.0559)	***
Meat industry	1.002 (1.0012–1.0019)	***	1.001 (1.0006–1.0014)	***	0.995 (0.9926–0.9965)	***	0.992 (0.9900–0.9938)	***
Fish industry	0.993 (0.9911–0.9951)	***	0.982 (0.9799–0.9840)	***	0.964 (0.9536–0.9755)	***	0.929 (0.9177–0.9412)	***
Vegetable industry	0.988 (0.9867–0.9885)	***	0.985 (0.9839–0.9856)	***	0.941 (0.9340–0.9478)	***	0.923 (0.9154–0.9300)	***
Animal oils and fats	0.982 (0.9812–0.9836)	***	0.980 (0.9789–0.9813)	***	0.909 (0.8988–0.9189)	***	0.888 (0.8781–0.8991)	***
Milk products	1.000 (0.9982–1.0013)	-	1.001 (0.9995–1.0024)	-	0.973 (0.9650–0.9806)	***	0.975 (0.9675–0.9822)	***
Grain mill industry	0.948 (0.9441–0.9523)	***	0.944 (0.9397–0.9478)	***	0.777 (0.7502–0.8047)	***	0.753 (0.7266–0.7811)	***
Bakery industry	0.984 (0.9809–0.9873)	***	0.977 (0.9740–0.9801)	***	0.974 (0.9589–0.9891)	**	0.938 (0.9236–0.9517)	***
Other food products	0.984 (0.9829–0.9861)	***	0.977 (0.9752–0.9783)	***	0.933 (0.9244–0.9412)	***	0.910 (0.9019–0.9178)	***
Animal feeding	0.998 (0.9967–0.9994)	**	0.999 (0.9975–1.0001)	-	0.970 (0.9630–0.9768)	***	0.967 (0.9605–0.9739)	***
Beverage industry	0.999 (0.9994–0.9996)	***	0.999 (0.9994–0.9996)	***	0.998 (0.9970–0.9983)	***	0.997 (0.9963–0.9978)	***
Forest industry	1.004 (1.0011–1.0077)	*	0.990 (0.9869–0.9931)	***	0.945 (0.9278–0.9632)	***	0.907 (0.8911–0.9240)	***
Leather and fur industry	1.070 (1.0624–1.0779)	***	1.078 (1.0702–1.0856)	***	1.110 (1.0776–1.1441)	***	1.115 (1.0823–1.1489)	***
Garden industry	0.922 (0.9122–0.9329)	***	0.922 (0.9119–0.9321)	***	0.717 (0.6715–0.7649)	***	0.709 (0.6649–0.7560)	***
Land use and cover								
ilr-Urban areas	1.006 (1.0048–1.0076)	***	1.013 (1.0114–1.0136)	***	1.050 (1.0440–1.0569)	***	1.062 (1.0566–1.0669)	***
ilr-Industrial areas	0.990 (0.9884–0.9921)	***	0.991 (0.9892–0.9926)	***	1.039 (1.0304–1.0477)	***	1.036 (1.0281–1.0442)	***
ilr-Agricultural areas	0.982 (0.9806–0.9835)	***	0.977 (0.9762–0.9786)	***	0.936 (0.9303–0.9422)	***	0.925 (0.9200–0.9300)	***
ilr-Forested areas	1.014 (1.0131–1.0158)	***	1.012 (1.0111–1.0136)	***	0.991 (0.9856–0.9971)	**	0.987 (0.9816–0.9925)	***

- non-statistically significant; * *p*-value < 0.05; ** *p*-value < 0.005; *** *p*-value < 0.0005.

## Data Availability

3rd Party Data. Restrictions apply to the availability of these data. Data was obtained from the RSAcovid19 records and are available with the permission of the Catalan Health Department.

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
