# Peer review of "Community Risk Factors in the COVID-19 Incidence and Mortality in Catalonia (Spain). A Population-Based Study"

_ijerph, 2021, doi:10.3390/ijerph18073768_

Round 1

Reviewer 1 Report

This study assessed the impacts of varying risk factors on the COVID19 illness and mortality. The risk factors include sociodemographic factors, underlying health conditions, and air pollution. This manuscript well focused the journal's aims, and will gain interests of this journal audiences. However, there are several points to be addressed. I recommend major revision. Below are detailed comments. 

Line 87: In the method section, the authors collected various data from different sources. A major concern is that all the data are historical 

Line 134: Why the authors selected PM10 instead of PM2.5? PM2.5 are known to affect immune responses because they deposit in deep lungs (i.e., alveoli), while PM10 stays in upper respiratory region.

Line 210: The authors need to explain better how to divide the low/medium/high category.

Line 258: The Table 4 shows same results with Figure 1. It can be moved to the supplementary data.

Line 329: The figure labels and texts are too small. Larger text sizes will help the audiences.

Line 445-447: This line of industry (e.g., any industry has high working density, virus-liked environments, etc.) could be named as "prone-to-infection industry" to have broader impacts. In this way, future research using the similar model can replace the "meat industry" variable to any types of "prone-to-infection industry" to predict COVID19 cases.

Line 537-542: This limitation could be mentioned in the method section to explain why the authors collected historical data (i.e., land use in 2017, SES/disease history in 2014) rather than the data in 2020 to explain better COVID19 situation.

Line 547: Why the population density was not included in the analysis?
As mentioned, the population density would be an important factor to explain COVID19 cased.

Author Response

Dear Reviewer,

We are so grateful to receiving your comments and evaluations. They have allowed us to revise some statements and make them clearer and more accurate.  

Please, find attached the responses to all your suggestions.

Best regards,

Quim Zaldo-Aubanell

Line 87: In the method section, the authors collected various data from different sources. A major concern is that all the data are historical 

Thanks for the appraisal. We have made it more clear in the limitations section [lines 590-593]. Given that we did not collected new data, we had to avail ourselves of pre-existent data. This made us unable to conduct our analyses with all the data referring to the same year. In this sense, although having all datasets from the same years might enhance the analyses, our results were consistent with previous literature, making them valid and adding evidence as to the independent effect of the assessed covariates. Parallelly, we did conduct some analyses to detected possible significant differences of more updated data (but less representative) and more historical for air pollutant concentration, and we did not detect any significant difference. For other covariates such as comorbidity, we were assessing illnesses that were prevalent. So, in general terms, we might assume that not huge difference can be observed from one year to another as to the pre-existent comorbidity of a given region.

Line 134: Why the authors selected PM10 instead of PM2.5? PM2.5 are known to affect immune responses because they deposit in deep lungs (i.e., alveoli), while PM10 stays in upper respiratory region.

We agree. It would have been of our will to be able to introduce O3 and PM2.5 to our models. As you suggest, their impact on health has been widely studied. We have now dedicated a paragraph addressing this fact in methodology section [lines 173-181] and limitations section [lines 587-589]. Given that our research analysed the entire Catalonia, we opted for using PM10 and NO2 as they were the only two air pollutants that were provided for all the Catalan territory and properly modelled.

Line 210: The authors need to explain better how to divide the low/medium/high category.

We agree. We have opted for a different analysis of the LULC data. We believe that, using a quantile division and performing the analysis with this variable as categorical made the understanding somewhat difficult to follow. We have opted for an isometric logratio transformation (ilr) that is suitable in Compositional data (like LULC data) [lines 218 - 232]. By conducting this transformation, we have been able to provide just one estimate for each LULC variable considered which enhance readability of the paper and results comprehension.

Line 258: The Table 4 shows same results with Figure 1. It can be moved to the supplementary data.

Completely agree. These two represented the same results. Given that we now show the unadjusted and adjusted estimates for each variable in Table 4, we have move Figure 1 (now Figure 6a) to supplementary data, as suggested.

Line 329: The figure labels and texts are too small. Larger text sizes will help the audiences.

We agree, the resolution was not suitable for a good readability. We have corrected this fact enlarging the text and labels of the figures.

Line 445-447: This line of industry (e.g., any industry has high working density, virus-liked environments, etc.) could be named as "prone-to-infection industry" to have broader impacts. In this way, future research using the similar model can replace the "meat industry" variable to any types of "prone-to-infection industry" to predict COVID19 cases.

Thanks for your interesting insight. We have now added a sentence [lines 504-506] addressing this suggestion.

Line 537-542: This limitation could be mentioned in the method section to explain why the authors collected historical data (i.e., land use in 2017, SES/disease history in 2014) rather than the data in 2020 to explain better COVID19 situation.

We agree with your comment. In each subsection of methodological section describing the variables considered, we have stated why we chose each dataset and clarified what they controlled for.

Line 547: Why the population density was not included in the analysis?
As mentioned, the population density would be an important factor to explain COVID19 cased.

Thank you for your evaluation. Although it is true that some research use population density as a covariate inside the model, we already accounted for its effect through our statistical analysis. In this regard, we fitted a generalized linear model, in the binomial family with a logit link. This methodology was used because our dependent variable was a binary (0 if there were no case or dead, and 1 if there were a case or dead). When calculated for a given region, the number of successes (ones) and failures (zeroes) are seen over the population of that area which gives us the incidence rates [lines 243-252]. In this sense, as we were fitting rates, we were already considering the impact of population to the total number of successes (people infected or dead). In addition, other assessed covariates in the model such as air pollutant or LULC date are strongly associated with population (more population live in more polluted areas, for instance).

Please notice that, given that the main model has been redesigned, Results and Discussion sections have been rewritten entirely.

Reviewer 2 Report

This manuscript describes application of statistical modeling to determine covariates of SARS-CoV-2 incidence and mortality in Catalonia. While I think it is laudable that the authors are trying to examine the role of factors such as land use, mobility, and industry in the incidence and mortality of SARS-CoV-2, I have concerns about the methodology.

  1. The statistical modeling applied here needs to be described in more detail. How exactly are comorbidities, demographics, socioeconomic status, and other known factors being controlled for? What software was used to implement and fit the model?
  2. I am concerned that the authors are considering too many possible factors without enough data to support the model fitting. This could lead to spurious results. The authors themselves note several seemingly odd correlations, specifically:
    1. Why are both high and low bare land protective for incidence, while medium bare land has no effect? It would seem to me that if bare land had any effect, you would see an increasing or decreasing trend.
    2. Similarly, there is increased mortality with both high and low industry, but no effect for medium industry?
    3. Regions with high, medium, and low urban areas all have higher mortality. Regions with high, medium, and low agriculture all have lower mortality. All these results suggest to me that land use doesn't seem to have much impact on incidence or mortality.
    4. There is also no consistent trend for both mortality and incidence for socioeconomic class.
    5. Hypertension and diabetes appear to lower incidence and mortality.

While the authors try to explain this last point as "collider bias", the fact that there are so many odd results suggests to me that too many factors are being considered with too little data to support the findings.

Author Response

Dear Reviewer,

We really appreciate your time and effort to examine our research and make such relevant comments. We have addressed them all and we strongly believe that they have substantially enhanced our work.

Hereunder, we provide answers to all your comments.

Best regards,

Quim Zaldo-Aubanell

  1. The statistical modeling applied here needs to be described in more detail. How exactly are comorbidities, demographics, socioeconomic status, and other known factors being controlled for? What software was used to implement and fit the model?

We agree with your comment. We have included some sentences in the methods section describing more in-depth the used covariates, and have explained what they controlled for. In a nutshell, comorbidity variables allowed us to control for the general health status of the population dwelling in each Basic Health Area (our unit of analysis). Demographic variables (sex and age) allowed us to analyze the impact between sexes and to properly account for the percentage of people older than 65 years. Finally, the non-clear debate as to the effect of socioeconomic characteristics for population-based studies motivated us to consider this information, as well. All variables had been previously described as important predictors of both COVID-19 incidence and mortality and, in order to further assess the effect of air pollutants, type of agri-food industry, and LULC data, we believed the model needed to control for this relevant information and detect their effect.

As for the concern about the software used, we now have added a sentence in the statistical analysis subsection (Methods section) in which we state “Statistical analysis was conducted using the R language environment for statistical computing, R version 3.6.2 (2019-12-12)”. Besides, the proper citation has also provided.

  1. I am concerned that the authors are considering too many possible factors without enough data to support the model fitting. This could lead to spurious results.
    Thanks for your evaluation. We did not detect problems related to the number of factors we analyzed. In fact, we were building our analysis with much data to allow for statistical inference. And if we would have problems related to the degrees of freedom of our analysis, we believe this might led to inferences problems, not detecting statistically significant associations, for instance. However, regarding the concern you expose, we agree that our previous proposal led us to some results that were difficult to communicate. In this sense, as suggested also by Reviewer 3, we have redesigned the main model which allowed us to be clearer on our statements and results.

Instead of considering some relevant comorbidities namely neoplasia, dementia, diabetes, congestive heart failure, hypertension, and ischemic cardiomyopathy, we have considered the following three group variables: cardiovascular diseases, psychologic disorders, and all-cause cancer. We believe that this new design better represents the general health status of the population, considering relevant illnesses that have previously described in the literature (we expose them in more detail in lines 119-135). Moreover, this new design allows us to easily interpret our estimates without limitations like collider bias.

The authors themselves note several seemingly odd correlations, specifically:

    1. Why are both high and low bare land protective for incidence, while medium bare land has no effect? It would seem to me that if bare land had any effect, you would see an increasing or decreasing trend.

We agree with your appraisal. Bare land represents a very little portion of the total land use and cover of Catalonia. As done previously with waterbodies, we have considered not taking it into account in our analyses. Furthermore, the four LULC categories left (urban, industrial, agricultural and forested) are much easier to relate with pollutant concentrations and thus, derive more understandable results.  

    1. Similarly, there is increased mortality with both high and low industry, but no effect for medium industry?
    2. Regions with high, medium, and low urban areas all have higher mortality. Regions with high, medium, and low agriculture all have lower mortality. All these results suggest to me that land use doesn't seem to have much impact on incidence or mortality.

Thanks for your considerations. We agree that using LULC data as a categorical variable with 4 levels (low, medium, high, and very high) did not result in clear and communicative results. Taking into consideration your point, we have analyzed LULC data through an isometric logratio transformation (ilr) for Compositional data [lines 218-232]. This transformation has allowed us to detect the contribution of each LULC category on both COVID-19 incidence and mortality providing just one coefficient for each LULC variable (instead of the 3 shown previously when compared to the “low” category). In this sense, we have reported that urban and industrial areas are positively associated with COVID-19 mortality while agricultural and forested areas are negatively associated, for instance.

We believe that this change has contributed to better readability and overall comprehension of the paper. 

We really appreciate your observations that have led us to change the methodology to analyze LULC data, making the overall paper more coherent and easier to follow.

    1. There is also no consistent trend for both mortality and incidence for socioeconomic class.

We agree. We have developed this question further in the discussion section, and provide more references [lines 405-430]. As you may have noticed in our results, we report that contrarily to other regions (UK or US, for instance), the COVID-19 incidence and mortality are associated with higher SES class. This is consistent with other research conducted in Catalonia (Saez et al., 2020) and other parts of Spain (Paez et al.,2020). Some authors even highlight that mobility is significantly and positively associated with higher SES in Girona region (East of Catalonia) while is significantly and negatively associated with lower SES in Barcelona (Ribas et al., 2021). In this sense, we suggest that, for a better understanding of the impact of SES data, individual-level based SES data needs to be accounted for in future research.

    1. Hypertension and diabetes appear to lower incidence and mortality.

Totally agree. As described above in question 2, we have solved interpretation problems like this with the new proposed model.

While the authors try to explain this last point as "collider bias", the fact that there are so many odd results suggests to me that too many factors are being considered with too little data to support the findings.

Thank you very much for your insightful thoughts. The awareness you have exposed in this regard has allowed us to rethink and redesign the core of the study which was our main model. We strongly believe that this new proposal is much robust, providing results that are aligned and comparable with previous relevant research.

Please notice that, given that the main model has been redesigned, the Results and Discussion sections have been rewritten entirely.

Reviewer 3 Report

The manuscript investigates the associations between both COVID-19 incidence and mortality and various baseline risk factors including air pollutant concentration, traffic intensity, type of agrifood industry, and overall land use and cover in conjunction with demographic and socioeconomic data. A generalized linear binomial model with logit link is used to assess the association between the incidence and mortality rate of COVID-19 and the selected covariates while adjusting for demographics, socioeconomic and comorbidity covariates. In general, this manuscript is well organized and presents interesting analysis and results. However, as the authors already pointed out, the input data used were rather out-of-date and scattered over multiple years from 2014 to 2020, and that may have resulted in contradicting findings. To my understanding, air pollutant concentration should be one of the most important risk factors this research is trying to address, while other factors such as agrifood and land use and cover should be somewhat underlying factors being related to air quality. But the analysis on how these factors relate to air quality is very limited. In addition to NO2 and PM10 (PM2.5 would be a better indicator than PM10, because it can be breathed into the respiratory system), to fully examine the impact of air quality, another important criteria pollutant, ozone, should also be included because its direct impact on respiratory and cardiovascular diseases that are also the primary health concerns caused by COVID-19. Since the authors are trying to link the transmissible ability with air quality (specifically aerosol) as alluded from Lines 479-484, the air quality data should be collected during the pandemic. Besides differing meteorological conditions, as it is well known that the COVID lockdowns and social distancing have also significantly altered air pollutant levels and compositions, therefore, the prior year’s data can’t be used to represent the air quality conditions under COVID-19.

Other confusing points include the reduced risk of both COVID-19 incidence and mortality caused by diabetes and hypertension. Even though the authors have pointed out that these findings are contrary to recent literature, they are also contrary to the common intuition. At most, these diseases don’t aggravate COVID-19 incidence and mortality, but it is impossible for these diseases to help reduce the incidence and mortality significantly, and this makes one to suspect either the experiment design or the misuse of the input data.  The authors should carefully redesign their study and perform more in-depth analysis before this research work can be published.

Specific comments:

  1. Lines 142-160, now that after controlling for possible differences between both periods (2016 and 2018/2019), the NO2 an PM10 annual average for 2016 was chosen, I don’t understand why the data for the period 2018/2019 was further discussed for the next three bullets. It is not clear that which period of the annual weighed average of NO2 and PM10 are presented in Figures 3a and 4a?
  2. Lines 218-219, this sentence is completely out of place and it goes nowhere. It seems that this sentence belongs to somewhere below section 3 Results after Line 242.
  3. Lines 228 to 229, the symbols are messed up, and in addition, since linear binomial model is the backbone of this research, more description should be provided, such as how the adjustment is done for demographics, socioeconomic and comorbidity covariates.
  4. Line 366, “have not experimented this phenomenon” should be “have not experienced this phenomenon”.
  5. Line 512, “which we have not been able to detect with once we grouped them”, don’t understand what does this mean?
  6. Figures 3a and 4a, the units are not right. It should be (µg/m3) and please specify which period this data belongs to.

Author Response

Dear Reviewer,

Thank you very much for all your interesting and insightful comments and suggestions. We believe that they have contributed a great deal to enhancing our research and making it easier to communicate.

Please, find below all our responses to your petitions.

Best regards,

Quim Zaldo-Aubanell.

However, as the authors already pointed out, the input data used were rather out-of-date and scattered over multiple years from 2014 to 2020, and that may have resulted in contradicting findings.

Thanks for your comment. We have made sure to include a sentence in each subsection of the Methods section explaining why we were using each dataset. As we did not collect new data, we had to avail ourselves of pre-existent, accessible, and reliable data. Although this could have limited our study (and we mention this in Limitation section lines 590-593), it is also true that we have controlled for possible significant differences between air pollutant datasets. Moreover, as to the comorbidity data, we were using prevalent illnesses that rarely change their distribution from one year to another. Nevertheless, our findings are aligned with previous research, suggesting we were able to detect the independent effect of the assessed covariables on both COVID-19 incidence and mortality.

To my understanding, air pollutant concentration should be one of the most important risk factors this research is trying to address, while other factors such as agrifood and land use and cover should be somewhat underlying factors being related to air quality. But the analysis on how these factors relate to air quality is very limited

Thanks for your comment. We agree. We have included some relevant references that analyzed the relationship between air pollution and LULC data (references 29 to 35, lines 70-74). Although we indeed have the data and have conducted a regression analysis to verify such associations, we believed that the best way to state such a relationship was through the relevant references.

In addition to NO2 and PM10 (PM2.5 would be a better indicator than PM10, because it can be breathed into the respiratory system), to fully examine the impact of air quality, another important criteria pollutant, ozone, should also be included because its direct impact on respiratory and cardiovascular diseases that are also the primary health concerns caused by COVID-19.

Completely agree. It would have been of our interest to be able to introduce O3 and PM2.5 to our models. As you suggest, their impact on health has been widely studied. Please find the paragraph that we now dedicate to addressing this fact in the methodology section [lines 173-182] and limitations section [lines 587-589]. Given that our research analyzed the entire Catalonia, we opted for using PM10 and NO2 as they were the only two air pollutants that were provided for all the Basic health areas (our unit of analysis) properly modeled.

Since the authors are trying to link the transmissible ability with air quality (specifically aerosol) as alluded from Lines 479-484, the air quality data should be collected during the pandemic. Besides differing meteorological conditions, as it is well known that the COVID lockdowns and social distancing have also significantly altered air pollutant levels and compositions, therefore, the prior year’s data can’t be used to represent the air quality conditions under COVID-19.

Totally agree. We believe that the pathway linking more impact of COVID-19 and air pollution is a worse general health status of the population. We mention the fact that some research is addressing the potential air pollution-to-human transmission (e.g. airborne transmission). However, as suggested, we stress the fact that both the data used and the model conducted make us unable to study this hypothesis [lines 462-472]

Other confusing points include the reduced risk of both COVID-19 incidence and mortality caused by diabetes and hypertension. Even though the authors have pointed out that these findings are contrary to recent literature, they are also contrary to the common intuition. At most, these diseases don’t aggravate COVID-19 incidence and mortality, but it is impossible for these diseases to help reduce the incidence and mortality significantly, and this makes one to suspect either the experiment design or the misuse of the input data.  The authors should carefully redesign their study and perform more in-depth analysis before this research work can be published.

We agree with your comment. As you mention, the results derived from our previous model were not easy to communicate. We have considered your proposal in depth and redesign our study to build a new main model. Instead of some relevant comorbidities namely neoplasia, dementia, diabetes, congestive heart failure, hypertension, and ischemic cardiomyopathy, we have considered the following three group variables: cardiovascular diseases, psychologic disorders, and all-cause cancer. We believe that this new design better represents the general health status of the population, considering relevant illnesses that have previously described in the literature (we expose them in more detail in lines 119-135). Moreover, this new design allows us to easily interpret our estimates without prior limitations such as collider bias.

We want to especially thank your contribution as we strongly believe that the new proposed main model helps better interpret the coefficients, enhancing the readability of the paper and overall comprehension.

Specific comments:

  1. Lines 142-160, now that after controlling for possible differences between both periods (2016 and 2018/2019), the NO2 an PM10 annual average for 2016 was chosen, I don’t understand why the data for the period 2018/2019 was further discussed for the next three bullets

We agree, there was no point in further examination of the non-elected dataset. Instead, we have moved the relevant information in the above lines [lines 162-165]

It is not clear that which period of the annual weighed average of NO2 and PM10 are presented in Figures 3a and 4a?

Thanks for your comment, we have made it clear in the caption of each map.

  1. Lines 218-219, this sentence is completely out of place and it goes nowhere. It seems that this sentence belongs to somewhere below section 3 Results after Line 242.

Thank you very much. We agree. We have moved this information to lines 169-172, after describing air pollutant information in the Methods section.

  1. Lines 228 to 229, the symbols are messed up, and in addition, since linear binomial model is the backbone of this research, more description should be provided, such as how the adjustment is done for demographics, socioeconomic and comorbidity covariates.

Completely agree. When fitting the article in the template provided, somehow the symbols messed up. We have made clear that this time they are appearing as they should. Regarding your thoughts about covariate data. More information has been provided in each subsection of the Methods section, accordingly.

  1. Line 366, “have not experimented this phenomenon” should be “have not experienced this phenomenon”.

We agree. This typographical mistake has been corrected.

  1. Line 512, “which we have not been able to detect with once we grouped them”, don’t understand what does this mean?

As we have rewritten the Discussion section, this sentence has been eliminated.

  1. Figures 3a and 4a, the units are not right. It should be (µg/m3) and please specify which period this data belongs to.

Thanks for the observation. This error has been properly addressed.

Please notice that, given that the main model has been redesigned, the Results and Discussion sections have been rewritten entirely.

Round 2

Reviewer 1 Report

The authors well addressed reviewer's comments. I recommend 'accept in present form' without further comment.

Author Response

Thanks for your comments.

We appreciate our revised document suited you.

Best regards,

Quim Zaldo-Aubanell

Reviewer 2 Report

The authors have addressed my comments to my satisfaction.

Author Response

(The authors gave the same response as above.)

Reviewer 3 Report

This revised manuscript has greatly improved by addressing all the reviewers' concerns, comments, and suggestions. Due to the lack of consistent and concurrent data inputs, the scientific strength and significance of results are compromised, but as an initial effort to address these complicated interconnections between COVID-19 incidence and mortality and air pollution, demographic, socioeconomic, comorbidity covariates, land use and land type, it has the merit to be published. 

Author Response

(The authors gave the same response as above.)
